# Characterization of *Streptomyces piniterrae* sp. nov. and Identification of the Putative Gene Cluster Encoding the Biosynthesis of Heliquinomycins

**DOI:** 10.3390/microorganisms8040495

**Published:** 2020-03-31

**Authors:** Xiaoxin Zhuang, Zhiyan Wang, Chenghui Peng, Can Su, Congting Gao, Yongjiang Wang, Shengxiong Huang, Chongxi Liu

**Affiliations:** 1Key Laboratory of Agricultural Microbiology of Heilongjiang Province, Northeast Agricultural University, Harbin 150030, China; zxx221661@163.com (X.Z.); ppoapeng@163.com (C.P.); g2456692645@163.com (C.G.); 2State Key Laboratory of Phytochemistry and Plant Resources in West China, Kunming Institute of Botany, Chinese Academy of Sciences, Kunming 650201, China; zhiyan_w@163.com (Z.W.); sucan126@126.com (C.S.); wangyongjiang@mail.kib.ac.cn (Y.W.)

**Keywords:** *Streptomyces**piniterrae* sp. nov., heliquinomycins, *Pinus yunnanensis*, genome analysis, biosynthetic gene cluster

## Abstract

A novel actinomycete producing heliquinomycin and 9’-methoxy-heliquinomycin, designated strain jys28^T^, was isolated from rhizosphere soil of *Pinus yunnanensis* and characterized using a polyphasic approach. The strain had morphological characteristics and chemotaxonomic properties identical to those of members of the genus *Streptomyces*. It formed spiral chains of spores with spiny surfaces. The menaquinones detected were MK-9(H_6_), MK-9(H_8_) and MK-9(H_4_). The major fatty acids were iso-C_16:0_, C_15:0_, C_16:1_ω7с and anteiso-C_15:0_. The phospholipids were diphosphatidylglycerol, phosphatidylmethylethanolamine, phosphatidylethanolamine and phosphatidylinositol mannoside. The DNA G + C content of the draft genome sequence, consisting of 8.5 Mbp, was 70.6%. Analysis of the 16S rRNA gene sequence showed that strain jys28^T^ belongs to the genus *Streptomyces* with the highest sequence similarities to *Streptomyces chattanoogensis* NBRC 13058^T^ (99.2%) and *Streptomyces lydicus* DSM 40002^T^ (99.2%) and phylogenetically clustered with them. Multilocus sequence analysis based on five other house-keeping genes (*atp*D, *gyr*B, *rpo*B, *rec*A and *trp*B*)* and the low level of DNA–DNA relatedness and phenotypic differences allowed the novel isolate to be differentiated from its most closely related strains. Therefore, the strain is concluded to represent a novel species of the genus *Streptomyces*, for which the name *Streptomyces*
*piniterrae* sp. nov. is proposed. Furthermore, the putative biosynthetic gene cluster of heliquinomycins was identified and the biosynthetic pathway was discussed. The type strain is jys28^T^ (=CCTCC AA 2018051^T^ =DSM 109823^T^).

## 1. Introduction

Actinobacteria has been the most fruitful source of microorganisms for all types of bioactive metabolites, including antibiotics, immunosuppressive agents, antitumor agents, and enzymes [1]. Actinobacteria belonging to the genus *Streptomyces*, in particular, are excellent producers. Multiple structural types of antibiotics, including amino glycosides, chloramphenicol, tetracyclines, macrolides and β-lactams, have been isolated from cultures of this genus [2]. During our continuous efforts to discover new or bioactive natural products from actinobacteria, we have reported the chemical studies of *Streptomyces piniterrae* jys28^T^ isolated from rhizosphere soil of *Pinus yunnanensis*, and identified heliquinomycin and its new analogue, 9’-methoxy-heliquinomycin [3]. 

Since the initial isolation and discovery of rubromycins in 1953 from *Streptomyces collinus*, more abundant compounds of rubromycin family, including purpuromycin, griseorhodins and heliquinomycins, have been identified by the end of the 20th century [4,5,6,7]. Multitude of them exhibit different activities on microbial inhibition, cytotoxicity, DNA helicases inhibition, telomerase inhibition or HIV reverse transcriptase inhibition, which implied the possibility of rubromycin family as potential medicinal chemistry precursors [8,9,10]. Besides the diverse bioactivity, the complex intriguing spiroketal core of rubromycins has attracted the attention of several research groups for chemical synthesis and biosynthesis studies. Several total synthesis strategies have been successfully established for rubromycin compounds, whereas, the biosynthetic conversions of rubromycins still exist a lot uncertainties [11,12,13,14,15]. With the clear recognition of biosynthetic gene clusters of rubromycins, further studies of the individual enzymes are required to determine the exact biosynthetic pathway.

In this study, we investigated the taxonomic status of strain jys28^T^ using a polyphasic approach and surveyed the biosynthetic gene cluster of heliquinomycins in the genome. 

## 2. Materials and Methods

### 2.1. Strains

Strain jys28^T^ was isolated from rhizosphere soil of *Pinus yunnanensis* collected from Yuxi, Yunnan Province, southwest China (24°21′ N, 102° 32′ E). The rhizosphere soil sample was air-dried for 14 days at room temperature, suspended in sterile distilled water followed by a standard serial dilution technique and spread on dulcitol-proline agar (DPA) [16] supplemented with cycloheximide (50 mg L^−1^) and nalidixic acid (20 mg L^−1^). After 21 days of aerobic incubation at 28 °C, colonies were transferred and purified on oatmeal agar [International *Streptomyces* Project (ISP) medium 3] [17] and maintained as glycerol suspensions (20%, *v*/*v*) at −80 °C. The reference strains, *Streptomyces chattanoogensis* NBRC 13058^T^ and *Streptomyces lydicus* DSM 40002^T^, were purchased from the Deutsche Sammlung von Mikroorganismen und Zellkulturen (DSMZ) for comparative analysis.

### 2.2. Phenotypic Characterization

Spore morphology was observed using scanning electron microscopy (Hitachi SU8010, Hitachi Co., Tokyo, Japan) after cultivation on ISP 3 medium at 28 °C for 4 weeks. Cultural characteristics were determined after 2 weeks at 28 °C using ISP media 1–7 [17], modified Bennett’s agar (MBA) [18], Nutrient agar (NA) [19], and Czapek’s agar (CA) [19]. Growth at different temperatures (4, 10, 15, 18, 25, 28, 35, 37, 40 and 45 °C) was determined on ISP 3 agar after incubation for 2 weeks. Tolerance of pH range (pH 4–11, at intervals of 1 pH units), using buffer system described by Zhao et al. [20] and NaCl tolerance (0–13%, with an interval of 1 %, *w*/*v*) for growth were determined after 2 weeks growth in ISP 2 broth in shake flasks (250 rpm) at 28 °C. The utilization of sole carbon and nitrogen sources at a final concentration of 0.5% (*w*/*v*) was tested using ISP 9 agar as the basal medium following the methods of Gordon et al. [21]. Other physiological and biochemical characteristics were conducted based on a previous report [22].

### 2.3. Chemotaxonomic Characterization

Biomass for chemical studies was prepared by growing strain jys28^T^ in ISP 2 broth in shake flasks at 28 °C for 5 days. The isomers of diaminopimelic acid (DAP) in the cell wall were derivatized according to McKerrow et al. [23], and analyzed by a HPLC method using an Agilent TC-C18 Column (250 × 4.6 mm i.d. 5 μm) [22]. The whole-cell sugars were analyzed according to the procedures developed by Lechevalier and Lechevalier [24]. Phospholipids in cells were examined by two-dimensional TLC (thin-layer chromatography, Qingdao Marine Chemical Inc., Qingdao, China) and identified using the method of Minnikin et al. [25]. Menaquinones were extracted from freeze-dried biomass and purified according to Collins [26] and analyzed by a HPLC-UV method as described previously [22]. Fatty acids were prepared and analyzed by GC-MS using the method of Zhuang et al. [27].

### 2.4. Phylogenetic Analysis

The 16S rRNA sequence was determined as described by Wang et al. [28], and similarities between strain jys28^T^ and other type strains of valid species were calculated based on pairwise alignment using the EzTaxon-e server (https://www.ezbiocloud.net/) [29]. Phylogenetic trees were constructed using the neighbour-joining [30] and the maximum likelihood [31] algorithms with Kimura’s two-parameter model [32] via molecular evolutionary genetics analysis (MEGA) software version 7.0 [33]. Phylogenetic relationships of strain jys28^T^ were also confirmed using sequences for five individual housekeeping genes (*rec*A, *gyr*B, *atp*D, *rpo*B and *trp*B) (2481 bp). The sequences of jys28^T^ were obtained from the whole genome. The sequences of each locus were aligned using MEGA 7.0 software and trimmed manually at the same position before being used for further analysis. Phylogenetic analysis was performed as described above.

### 2.5. Genome Analysis

Genomic DNA was extracted according to the lysozyme-sodium dodecyl sulfate-phenol/chloroform method [34]. Whole-genome sequencing was performed on HiSeq 2500 Sequencing System (Illumina, San Diego, CA, USA) according to the User Guide and assembling on MiSeq plateform [35]. Genome mining analysis was performed with antiSMASH (version 4.0, Blin K, Oxford, UK) [36]. The digital DNA-DNA hybridization (dDDH) and average nucleotide identity (ANI) values were determined between the draft genome sequences of strain jys28^T^ and *S. chattanoogensis* NBRC 13058^T^ (LGKG01000001) and *S. lydicus* DSM 40002^T^ (CP007699) online at http://ggdc.dsmz.de using the Genome-to-Genome Distance Calculation (GGDC 2.0) [37] and the ChunLab’s online ANI Calculator (www.ezbiocloud.net/tools/ani) [38], respectively.

## 3. Results and Discussion

### 3.1. Polyphasic Taxonomic Characterization of Strain jys28^T^

Identification using the EzTaxon-e server revealed that strain jys28^T^ belonged to the genus *Streptomyces* with the highest 16S rRNA gene sequence similarities to *S. chattanoogensis* NBRC 13058^T^ (99.2%) and *S. lydicus* DSM 40002^T^ (99.2%). 16S rRNA gene sequence similarities between strain jys28^T^ and other species of the genus *Streptomyces* were lower than 99.0%. Phylogenetic analysis based on the 16S rRNA gene sequences indicated that the isolate clustered with the above-mentioned strains in the neighbour-joining tree (Figure 1). The tree topology in this region was also supported by the maximum-likelihood (Appendix A). To further clarify the affiliation of strain jys28^T^ to its closely related strains, partial sequences of housekeeping genes including *atp*D, *gyr*B, *rec*A, *rpo*B and *trp*B were obtained, the accession numbers for the *Streptomyces* sequences used are given in Appendix A and phylogenetic trees based on the neighbour-joining and maximum-likelihood algorithms were reconstructed from the concatenated sequence alignment (2481 bp) of the five housekeeping genes. The multilocus sequence analysis (MLSA) trees exhibited the close association between strain jys28^T^ and *S. lydicus* DSM 40002^T^ (Figure 2 and Appendix A). Based on the phylogenetic trees and 16S rRNA gene similarities, the isolate was mostly related to *S. chattanoogensis* NBRC 13058^T^ and *S. lydicus* DSM 40002^T^. dDDH indicated that DNA–DNA relatedness between strain jys28^T^ and *S. chattanoogensis* NBRC 13058^T^ and *S. lydicus* DSM 40002^T^ were 27.6–32.5% and 27.5–32.4%, respectively, which are much lower than the cut-off point of 70% recommended for the assignment of bacteria strains to the same genomic species [39]. ANI values between jys28^T^ and the two type strains were 84.9% and 85.0%, respectively, whose values are also below the recommended threshold for species delineation (95–96%) [40]. In addition, The MLSA distances between the isolate and the two type strains were 0.057 and 0.044, respectively (Appendix A), which was well above the species level threshold of 0.007 considered to be the threshold for species determination [41].

Morphological observation of a 4-week-old culture of strain jys28^T^ growth on ISP 3 medium revealed that it has the typical characteristics of the members of the genus *Streptomyces* [42]. Aerial and substrate mycelia were well developed without fragmentation. Spiral spore chains were observed, and spores (0.5–0.6 × 0.8–0.9 μm) were spiny and non-motile (Figure 3). Good growth was observed on all tested media. The colours of aerial mycelium varied from pure white to dark gray, and those of the substrate mycelium varied from brilliant yellow to dark purplish red. Diffusible pigments were observed on ISP 2, ISP 3, ISP 5, ISP 6, ISP 7, NA and CA. The significant colour variations of aerial and substrate mycelium and production of diffusible pigments on different media are listed in Appendix A, which shows some morphological differences between the isolate and its closely related strains. Strain jys28^T^ could grow at 40 °C, while its closely related strains could not. The utilization of L-arabinose and L-threonine, and hydrolysis of aesculin could also distinguish it from its closely related strains. Most notably, strain jys28^T^ could be distinguished readily from its closely related strains on the basis of NaCl tolerance. Other physiological and biochemical characteristics of strain jys28^T^ compared with its closely related strains are listed in Table 1.

Chemotaxonomic analysis revealed that strain jys28^T^ exhibited characteristics which are typical of members of the genus *Streptomyces* [42]. It contained LL-diaminopimelic acid as the cell-wall diamino acid and whole-cell sugars included glucose and xylose. The phospholipid profile was consisted of diphosphatidylglycerol, phosphatidylmethylethanolamine, phosphatidylethanolamine and phosphatidylinositol mannoside (phospholipid type II) [44] (Appendix A). The menaquinones detected were MK-9(H_6_) (53.1 %), MK-9(H_8_) (23.9%) and MK-9(H_4_) (23.0%). The major cellular fatty acids (> 10%) were iso-C_16:0_ (25.3%), C_15:0_ (15.8%), C_16:1_ω7с (11.9%) and anteiso-C_15:0_ (11.7%) (Table 2). The fatty acid profile of strain jys28^T^ was evidently different from those of *S. chattanoogensis* NBRC 13058^T^ and *S. lydicus* DSM 40002^T^, such as the presence of saturated fatty acids (C_16:0_ and C_18:0_) as the major fatty acids and absence of most branched fatty acids (anteiso-C_15:0_, anteiso-C_17:0_ and iso-C_14:0_) in *S. chattanoogensis* NBRC 13058^T^, the presence of C_14:0_ and absence of C_17:0_ and iso-C_14:0_ in *S. lydicus* DSM 40002^T^.

Therefore, it is evident from the genotypic and phenotypic data that strain jys28^T^ represents a novel species of the genus *Streptomyces*, for which the name *Streptomyces piniterrae* sp. nov. is proposed.

### 3.2. Description of Streptomyces piniterrae sp. nov.

*Streptomyces piniterrae* (pi.ni.ter’ra.e. L. n. pinus name of a plant and also a botanical generic name (*Pinus*); L. n. *terra* soil; N.L. gen. n. *piniterrae* of soil of pinus, denoting the source of the type strain).

Gram-stain-positive and aerobic actinomycete that forms well-developed, branched substrate hyphae and aerial mycelium that differentiate into spiral spore chains with oval spores. The spore surface is spiny. The colour of aerial mass varies from pure white to dark gray. Diffusible pigments are observed on ISP 2, ISP3, ISP5, ISP6, ISP7, NA and CA. Growth is observed at temperatures between 15 and 40 °C, with an optimum temperature of 28 °C. Growth occurs at pH values between 4 and 8 (optimum pH 7) and 11% NaCl tolerance. It is positive for production of urease, negative for hydrolysis of starch, aesculin, Tweens (20, 40 and 80), production of H_2_S, peptonization and coagulation of milk, decomposition of cellulose, liquefaction of gelatin and reduction of nitrate. Dulcitol, D-fructose, D-galactose, D-glucose, *meso*-inositol, lactose, D-maltose, D-mannitol, D-mannose, D-raffinose, L-rhamnose, D-ribose, D-sorbitol and D-sucrose are utilized as sole carbon sources but not L-arabinose and D-xylose. L-Alanine, L-arginine, L-asparagine, L-aspartic acid, glycine, L-glutamic acid, L-glutamine, L-proline and L-tyrosine are utilized as sole nitrogen sources but not creatine, L-serine, L-threonine and L-tryptophan. The cell wall diamino acid is LL-diaminopimelic acid. Whole-cell sugars contain glucose and xylose. The menaquinones are MK-9(H_6_), MK-9(H_8_) and MK-9(H_4_). The phospholipid profile consists of diphosphatidylglycerol, phosphatidylmethylethanolamine, phosphatidylethanolamine and phosphatidylinositol mannoside. The major fatty acids (>10%) are iso-C_16:0_, C_15:0_, C_16:1_ω7с and anteiso-C_15:0_.

The type strain, jys28^T^ (=CCTCC AA 2018051^T^ =DSM 109823^T^), was isolated from rhizosphere soil of *P. yunnanensis* collected from Yuxi, Yunnan Province, southwest China. The DNA G+C content of the draft genome sequence of the type strain indicates a value of approximately 70.6 mol% for the species. The GenBank accession number for the 16S rRNA gene sequence and the draft genome sequence of the type strain are MH620762 and SUMB00000000, respectively.

### 3.3. Identification of Secondary Metabolic Biosynthetic Gene Clusters, Including the Putative BGC for Heliquinomycins

Sequencing of the genome produced an annotated genome size of approximately 8.5 Mbp (28 scaffolds). This draft genome sequence has been deposited at the GenBank/EMBL/DDBJ under the accession number SUMB00000000. The genome contains one linear chromosome with 7 rRNA operons, 67 tRNA genes and 7543 protein-coding genes (CDSs). Genome sequencing showed the DNA G + C content of strain jys28^T^ to be 70.6 mol%. AntiSMASH analysis led to the identification of 33 putative gene clusters in the genome of strain jys28^T^. Five of these clusters were identified belonging to the family of polyketide synthases (PKSs), including type I (three clusters), II (one clusters) and III (one cluster). Likewise, genome sequence analysis detected seven additional gene clusters comprising modular enzyme coding genes such as non-ribosomal peptide synthetase (NRPS, five clusters) and hybrid PKS I-NRPS genes (three clusters). Other gene clusters included terpene (seven clusters), butyrolactone (two clusters), siderophore (four clusters), lantipeptide (two clusters), bacteriocin (two clusters), ladderane (two clusters) and ectoine (one cluster).

Based on BLAST sequence analysis, we deduced the biosynthetic gene cluster of heliquinomycins, which contained 10 polyketide core genes (*4027*–*4030*, *4033* and *4046*–*4050*), 21 tailoring genes (*4036*, *4038*–*4042*, *4051*, *4054*–*5058* and *4060*–*4068*), 2 regulatory genes (*4024* and *4025*), 1 resistance gene (*4059*) and 11 unassigned or unknown functional genes (*4026*, *4031*, *4032*, *4034*, *4035*, *4037*, *4043*–*4045*, *4052* and *4053*) (Appendix A). The gene cluster displayed high homology with that of griseorhodin A identified in *Streptomyces* sp. JP95 (Figure 4A) [45].

So far as is known, griseorhodin A was derived from iterative PKS pathway. Then, post-PKS modifications produce the intriguing 5, 6-spiroketal core through a highly complex oxidative tailoring process [46]. In line with griseorhodin A biosynthetic pathway, the carbon chain of heliquinomycin and 9’-methoxy-heliquinomycin was also elongated by mimic PKS and modified by putative 3-oxoacyl-ACP reductase (*4051* and/or *4056*) (Figure 4B) [13]. One of the difference between griseorhodin A and heliquinomycins is the methoxycarbonyl group at position C-25. Results of the comparison between *grh* and *hlq* showed an extra cytochrome P450 (*4054*) and an extra SAM-dependent methyltransferase (*4055*) in *hlq* gene cluster. The protein sequence of 4045 hits the conserved domain of *ent*-kaurene oxidase that suggests a function for 4045 in the three successive oxidations of C-26 [47]. Moreover, *rubU*, a homologous gene of *4054*, has been found in rubromycin (*rub*) biosynthetic gene cluster, which could characterize the role of *4054* from the side as rubromycin does have the same methoxycarbonyl group at position C-25. After the three-step-oxidation of C-26, a methyl group was introduced to the carboxyl group by putative methyltransferase 4040. Besides the methylation of this position, methyltransferase 4040 might also play a role at hydroxy group of A ring. The extra SAM-dependent methyltransferase (4055) was speculated to be responsible for the methoxy group of B ring as a result of the absent of this gene in both *grh* and *rub* clusters. Refer to the implied intermediates of rubromycins, methoxycarbonyl-collinone and methoxycarbonyl-lenticulone would probably be intermediates of heliquinomycins, which was tentatively verified by gene knockout experiment with the corresponding enzymes to FAD-dependent monooxygenase (4039) and GNAT family *N*-acetyltransferase (4037) [15,48]. The hydroxyl group on D ring showed another difference in structure between griseorhodin A and heliquinomycins. A relevant putative cytochrome P450 was present in both *grh* (*grhO3*) and *hlq* (4057) clusters, which might relate to the hydroxyl group or the epoxide in D ring of heliquinomycin and griseorhodin A, respectively. The proposed biosynthetic pathway for 2,6-dideoxy-3-*O*-methylhexopyranose sugar moiety of heliquinomycins was shown in Figure 4C. The gene cluster of the pyranose in *hlq* is 75% similar to that of *ave* [49,50]. Under the action of putative glycosytransferase (4066), ultimately formed the heliquinomycins. In conclusion, the relatively definite gene clusters of these spiroketal compounds have already been found previously, but the individual mechanism and substrate specificities of those numerous interesting tailoring enzymes have not been identified yet. Therefore, additional studies need to be carried out.

## 4. Conclusions

A heliquinomycin-producing strain, jys28^T^, was isolated from the rhizosphere soil of *Pinus yunnanensis*. Morphological and chemotaxonomic features together with phylogenetic analysis suggested that strain jys28^T^ belonged to the genus *Streptomyces*. Physiological and biochemical characteristics combined with ANI and dDDH values clearly revealed that strain jys28^T^ was differentiated from its closely related strains. Based on the polyphasic taxonomic analysis, it is suggested that strain jys28^T^ represents a novel species of the genus *Streptomyces*, for which the name *Streptomyces piniterrae* sp. nov. is proposed. The type strain is jys28^T^ (=CCTCC AA 2018051^T^ =DSM 109823^T^). In addition, the biosynthetic pathway and gene cluster of heliquinomycins were deduced by whole-genome analysis.

## Figures and Tables

**Figure 1 microorganisms-08-00495-f001:**
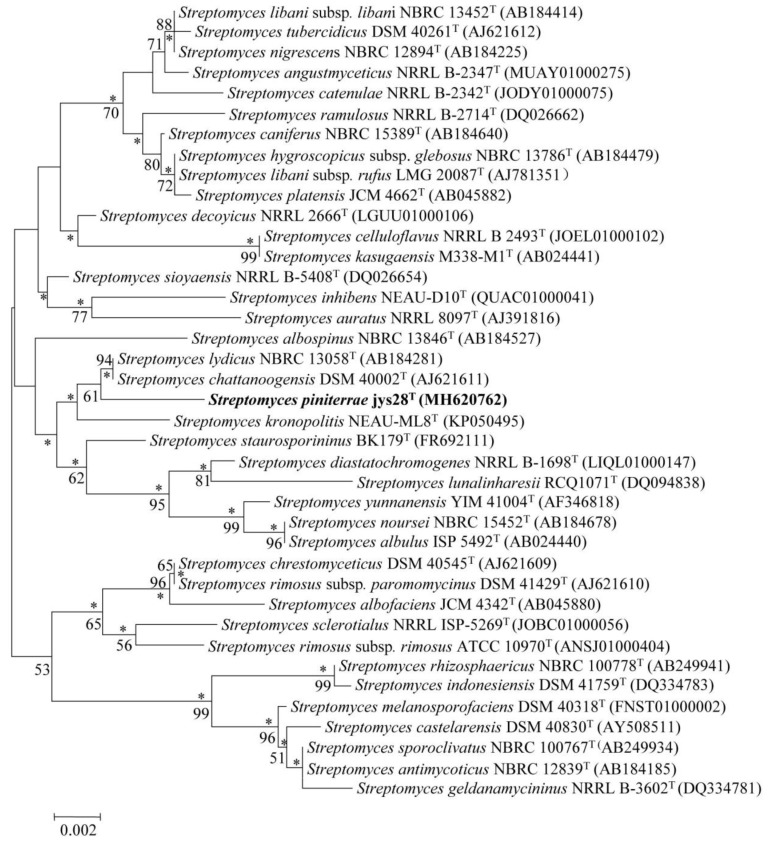
Neighbour-joining tree showing the phylogenetic position of strain jys28^T^ and related taxa based on 16S rRNA gene sequences. Asterisks indicate branches that were also found using the maximum-likelihood method. Numbers at branch points indicate bootstrap percentages (based on 1000 replicates); only values >50% are indicated. Bar, 0.002 substitutions per nucleotide position.

**Figure 2 microorganisms-08-00495-f002:**
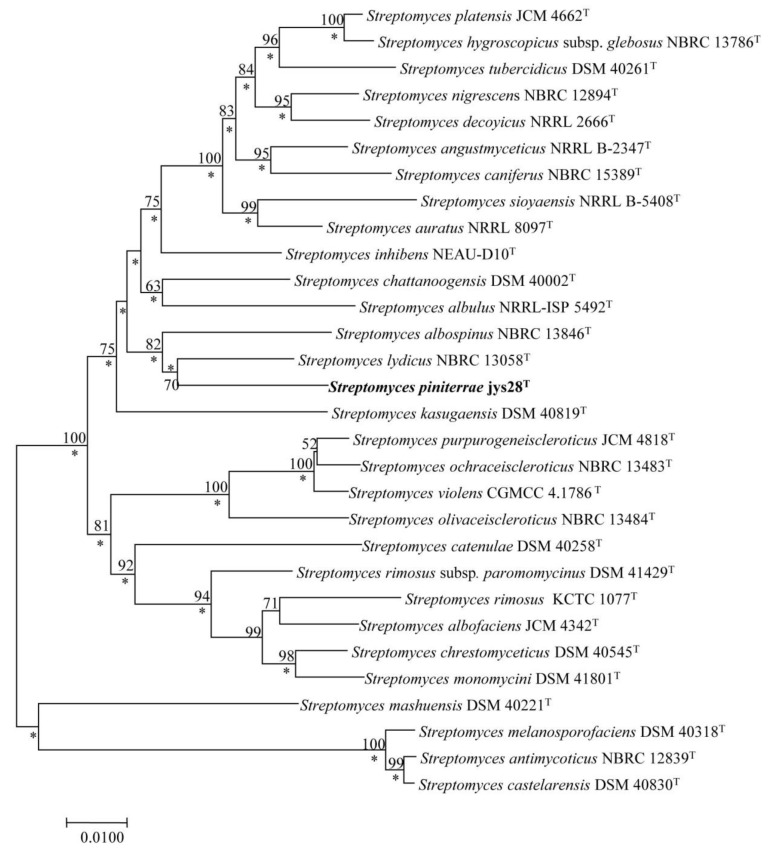
Neighbour-joining tree based on MLSA analysis of the concatenated partial sequences (2481 bp) from five housekeeping genes (*atp*D, *gyr*B, *rec*A, *rpo*B and *trp*B) of strain jys28^T^ and related taxa. Only bootstrap values above 50% (percentages of 1000 replications) are indicated. Asterisks indicate branches also recovered in the maximum-likelihood tree. Bar, 0.01 nucleotide substitutions per site.

**Figure 3 microorganisms-08-00495-f003:**
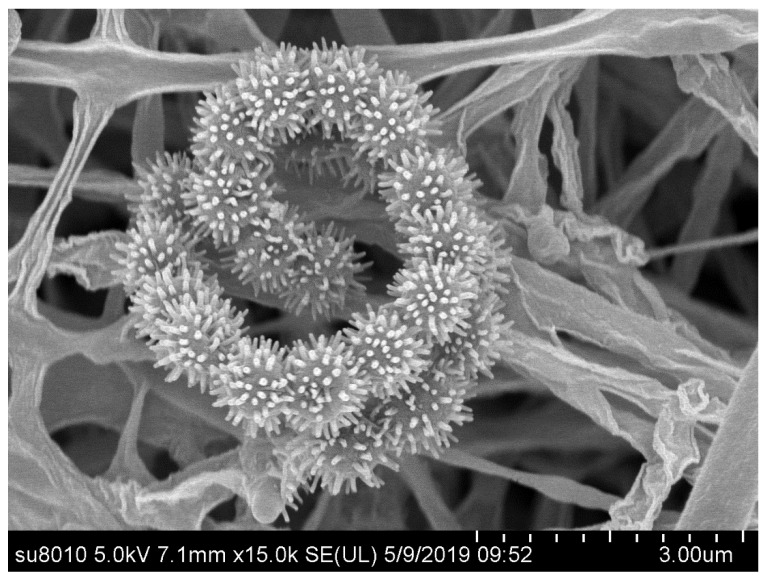
Scanning electron micrograph of strain jys28^T^ grown on ISP 3 agar for 4 weeks at 28 °C.

**Figure 4 microorganisms-08-00495-f004:**
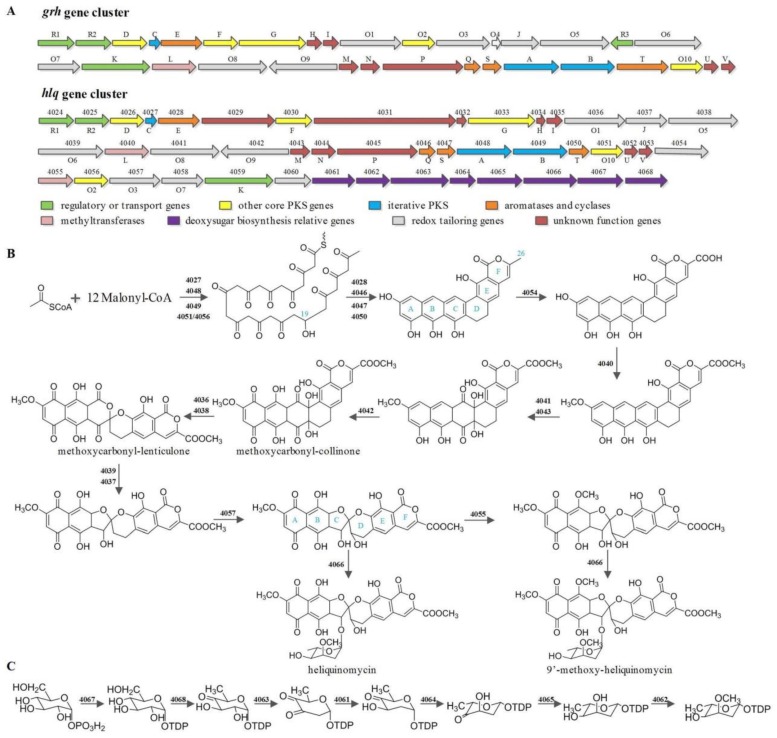
Identification of biosynthetic information of heliquinomycins. (**A**). Biosynthetic gene clusters of griseorhodin A (*grh*) and heliquinomycins (*hlq*). Individual genes are labeled by their numbers or names above the arrows. Genes of *grh* gene cluster which are homologous to those of *hlq* are exhibited under the arrows. (**B**). Proposed biosynthesis of the heliquinomycins. (**C**). Proposed biosynthesis of the 2,6-dideoxy-3-*O*-methylhexopyranose sugar moiety of heliquinomycins.

**Table 1 microorganisms-08-00495-t001:** Differential characteristics of strain jys28^T^ and its closely related strains. Strains: 1, jys28^T^; 2, *S. chattanoogensis* DSM 40002^T^; 3, *S. lydicus* NBRC 13058^T^. All data are from this study except as labeled. ^a^Data from Kim et al. [43]. +, positive; –, negative.

Characteristic	1	2	3
Spore surface	Spiny	Spiny^a^	Smooth^a^
Hydrolysis of aesculin	–	+	+
Production of H_2_S	–	+	–
Carbon source utilization			
L-arabinose	–	+	+
Lactose	+	–	+
L-rhamnose	+	–	+
D-ribose	+	–	+
D-xylose	–	–	+
Nitrogen source utilization			
L-aspartic acid	+	–	+
L-threonine	–	+	+
L-tryptophan	–	–	+
pH range for growth	4–8	5–9	5–9
Growth at 40 °C	+	–	–
Tolerance of NaCl (%, *w*/*v*)	11	2	2

**Table 2 microorganisms-08-00495-t002:** The cellular fatty acid compositions of strain jys28^T^ and its closely related strains. Strains: 1, jys28^T^; 2, *Streptomyces chattanoogensis* NBRC 13058^T^; 3, *Streptomyces lydicus* DSM 40002^T^. Values are percentages of total fatty acids. Fatty acids representing < 1% in all strains were omitted. All data are from this study, and all strains were cultivated in ISP 2 broth for 5 days under. the same conditions. –, not detected.

Fatty Acid	1	2	3
Saturated fatty acids			
C14:0	–	–	14.3
C15:0	15.8	–	18.2
C16:0	5.8	30.0	28.4
C_17:0_	6.9	1.7	–
C_18:0_	1.0	41.4	1.8
Unsaturated fatty acids			
C_16:1_ ω7c	11.9	16.2	2.8
C_17:1_ ω7c	7.0	–	1.3
C_18:1_ ω7c	2.6	7.9	1.9
Branched fatty acids			
anteiso-C_15:0_	11.7	–	5.6
anteiso-C_17:0_	8.9	–	11.1
iso-C_14:0_	3.1	–	–
iso-C_16:0_	25.3	1.1	12.7

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
