# Peer review of "Characterization of *Streptomyces piniterrae* sp. nov. and Identification of the Putative Gene Cluster Encoding the Biosynthesis of Heliquinomycins"

_microorganisms, 2020, doi:10.3390/microorganisms8040495_

Round 1

Reviewer 1 Report

The work of Zhuang et al. analyzes the morphological, partial biomass composition, and genome sequence of a streptomycete they isolated from rhizosphere soil.  Overall, the work is solid.  This type of study is very traditional in the field of Streptomyces research.

Major comments:

The authors report the genome sequence for this strain, jys28, and identify the putative biosynthetic gene cluster for heliquinomycin based on structural similarity to griseorhodin A through BLAST.  This is fine, but the authors should run their sequences through antiSMASH or a related algorithm to identify other potential clusters as well.

It’s not clear to me how the phospholipid, fatty acid, and menaquinone content was determined.  They seem to reference a previous study (ref 22), but when I looked that up, ref 22 didn’t seem to have any information either.  Referencing a prior paper is of course fine, but it feels insufficient here; the authors should provide more detail.  For example, I assume they used HPLC for much of their analysis, so describing their instrument settings would be a good idea. 

Minor comments:

Line 63: is ref 16 correct?  I looked it up and did not see any mention of dulcitol-proline agar

Line 96: please mention which technology was used for sequencing.  Was it Illumina, Pac Bio, Nanopore, or some combination of the three?  How was assembly done? 

Lines 144 – 146 are repeated in lines 183 – 185.  This is not necessary.

Paragraph starting at line 211 is nothing but speculation and should be shortened.  Perhaps highlight key differences between the heliquinomycin and griseorhodin A clusters and how they might impact the structures of the two molecules. 

3.3 heading is not correct because the authors did not actually analyze the whole genome; they simply identified the BGC for heliquinomycin.  Assuming the authors do an antiSMASH (or similar) analysis, a better heading would be something like, “Identification of secondary metabolic biosynthetic gene clusters, including the putative BGC for heliquinomycin.”

Likewise, their title is not quite correct.  A better title would be something like, “Characterization of Streptomyces piniterrae sp. nov. and Identification of the Putative Gene Cluster Encoding the Biosynthesis of Heliquinomycins.  In essence, it is not correct to say “whole genome analysis of a Gene Cluster” in English.  A BGC is a specific region of the genome, so one cannot do a “WHOLE genome analysis” of a specific region. 

Author Response

Answers to the comments of Reviewer #1

Thank you very much for your constructive suggestions.

  1. The authors report the genome sequence for this strain, jys28, and identify the putative biosynthetic gene cluster for heliquinomycin based on structural similarity to griseorhodin A through BLAST. This is fine, but the authors should run their sequences through antiSMASH or a related algorithm to identify other potential clusters as well.

We have run the genome through antiSMASH to identify other potential clusters (see lines 108, 221-228).

  1. It’s not clear to me how the phospholipid, fatty acid, and menaquinone content was determined. They seem to reference a previous study (ref 22), but when I looked that up, ref 22 didn’t seem to have any information either. Referencing a prior paper is of course fine, but it feels insufficient here; the authors should provide more detail. For example, I assume they used HPLC for much of their analysis, so describing their instrument settings would be a good idea.

We have described the methods for chemotaxonomic characterization (see lines 82-92).

  1. Line 63: is ref 16 correct? I looked it up and did not see any mention of dulcitol-proline agar

Reference 13 and 14 corresond to the same reference in previous manuscript. As a matter of fact, the ref 16 is correct. Please see the revised manuscript (ref 16).

  1. Line 96: please mention which technology was used for sequencing. Was it Illumina, Pac Bio, Nanopore, or some combination of the three? How was assembly done?

We have added the method in the revised manuscript (see lines 105-109)

  1. Lines 144 – 146 are repeated in lines 183 – 185. This is not necessary.

We think this part should be kept. Lines 150-156 is a description of the phenotypic characteristic results for strain jys28T, whereas lines 193-196 is a description of Streptomyces piniterrae sp. nov. (species).

  1. Paragraph starting at line 211 is nothing but speculation and should be shortened. Perhaps highlight key differences between the heliquinomycin and griseorhodin A clusters and how they might impact the structures of the two molecules.

We have revised according to your suggestion (see lines 237-239).

  1. 3.3 heading is not correct because the authors did not actually analyze the whole genome; they simply identified the BGC for heliquinomycin. Assuming the authors do an antiSMASH (or similar) analysis, a better heading would be something like, “Identification of secondary metabolic biosynthetic gene clusters, including the putative BGC for heliquinomycin.”

We have revised according to your suggestion (see lines 215-228).

  1. Likewise, their title is not quite correct. A better title would be something like, “Characterization of Streptomyces piniterrae sp. nov. and Identification of the Putative Gene Cluster Encoding the Biosynthesis of Heliquinomycins. In essence, it is not correct to say “whole genome analysis of a Gene Cluster” in English. A BGC is a specific region of the genome, so one cannot do a “WHOLE genome analysis” of a specific region.

We have revised according to your suggestion (see title).

Reviewer 2 Report

Comments and Suggestions for Authors

Review of the article entitled: “Characterization of Streptomyces piniterrae sp. nov. and Whole-Genome Analysis of The Gene Cluster Encoding the Biosynthesis of Heliquinomycins”

In this manuscript, the authors reported the characterization of a novel Streptomyces by polyphasic approach and the analysis of the gene cluster encoding the biosynthesis of heliquinomycins. All the experiments have been conducted logically and the experimental design is well-suited. However, I have some major concerns that I hope they could address in a revised version of the manuscript.

Title - in my opinion the authors did not analyse the whole-genome sequence, but after sequencing the whole genome they analyzed only the gene cluster for Heliquinomycin biosynthesis.

Abstract – the abstract is well prepared and informative. I do not have any critical remarks

Introduction –

lines 41-43 “we have reported the chemical studies of Streptomyces sp. jys28 isolated from rhizosphere soil of Pinus yunnanensis, and identified heliquinomycin and its new analogue, 9’-methoxy-heliquinomycin [3].” For me it is not clear if it is the same strain you are discussing about in the present manuscript.

Line 51 “synthesis and biosynthesis studies”, Do the authors mean chemical and biological ones?

Lines 53-55 please rephrase the sentence.

Materials and Methods 

Line 90 Since MLSA was used, and in the results paragraph you write partial sequences, I would suggest adding here the amplicon sizes.

Lines 95-96 “Genomic DNA extraction and whole-genome sequencing of strain jys28T were performed as described in a previous report [22].” Is the reference correct? 1974?

Results and Discussion –

The paragraph “3.2. Description of Streptomyces piniterrae sp. nov.” is a list of features not well reported. For example, some sentences lack of the subject.

The paragraph “3.3 Whole-Genome Analysis of The Biosynthesis of Heliquinomycins” is confusing.

First, I would suggest to describe the main results of WGA, reporting here the main results written in the paragraph 2.4 Genome analysis. For example, Sequencing of the genome produced an annotated genome size of approximately 8.5 Mbp. The genome contains one linear chromosome with 7 rRNA operons, 67 tRNA genes and 7543 protein-coding genes (CDSs). Then, I suggest to describe the gene cluster and finally the biosynthetic pathway.

Some typos should be corrected, i.e. line 208 resistant gene,

lines 213-214 the carbon chain of heliquinomycin and 9’-methoxy-heliquinomycin was also elongated,

line 230 of the absent of this gene,

line 230 in both grh and rub cluster,

line 231, biosynthesis genes,

lines 233-234 has been implied as intermediates or shunt products,

line 250, Biosynthesis gene clusters,

lines 257-258, together with phylogenetic analysis and genomes suggested.

References-13 and 14 correspond to the same reference.

Supporting material – Correct Table S1. GenBank accession numbers of the sequnces

Author Response

Answers to the comments of Reviewer #2

Thank you very much for your constructive suggestions.

  1. Title- in my opinion the authors did not analyse the whole-genome sequence, but after sequencing the whole genome they analyzed only the gene cluster for Heliquinomycin biosynthesis.

We have revised (see title).

  1. lines 41-43 “we have reported the chemical studies of Streptomyces sp. jys28 isolated from rhizosphere soil of Pinus yunnanensis, and identified heliquinomycin and its new analogue, 9’-methoxy-heliquinomycin [3].” For me it is not clear if it is the same strain you are discussing about in the present manuscript.

We have revised (see line 42).

  1. Line 51 “synthesis and biosynthesis studies”, Do the authors mean chemical and biological ones?

It should be chemical synthesis and biosynthesis. We have revised (see line 51).

  1. Lines 53-55 please rephrase the sentence.

We have revised (see lines 53-55).

  1. Line 90 Since MLSA was used, and in the results paragraph you write partial sequences, I would suggest adding here the amplicon sizes.

We have added (see line 100).

  1. Lines 95-96 “Genomic DNA extraction and whole-genome sequencing of strain jys28T were performed as described in a previous report [22].” Is the reference correct? 1974?

We have revised (see lines 105-107).

  1. The paragraph “3.2. Description of Streptomyces piniterrae sp. nov.” is a list of features not well reported. For example, some sentences lack of the subject.

We have revised (see lines 196-204).

  1. The paragraph “3.3 Whole-Genome Analysis of The Biosynthesis of Heliquinomycins” is confusing.

First, I would suggest to describe the main results of WGA, reporting here the main results written in the paragraph 2.4 Genome analysis. For example, Sequencing of the genome produced an annotated genome size of approximately 8.5 Mbp. The genome contains one linear chromosome with 7 rRNA operons, 67 tRNA genes and 7543 protein-coding genes (CDSs). Then, I suggest to describe the gene cluster and finally the biosynthetic pathway.

We have revised according to your suggestion (see lines 215-228).

  1. Some typos should be corrected, i.e. line 208 resistant gene, ines 213-214 the carbon chain of heliquinomycin and 9’-methoxy-heliquinomycin was also elongated,

line 230 of the absent of this gene,

line 230 in both grh and rub cluster,

line 231, biosynthesis genes,

lines 233-234 has been implied as intermediates or shunt products,

line 250, Biosynthesis gene clusters,

lines 257-258, together with phylogenetic analysis and genomes suggested.

We have revised according to your suggestion (see lines 232, 238, 245, 250 and 273).

  1. References-13 and 14 correspond to the same reference.

We have revised (see lines 314-318).

  1. Supporting material– Correct Table S1. GenBank accession numbers of the sequnces

We have revised (seeTable S1).

Round 2

Reviewer 2 Report

The authors addressed all the suggestions and comments I gave fot the first version of the ms.

Thus, I think the ms was improved and deserves to be published.

Best regards